# HIV-positive status disclosure and associated factors among children in public health facilities in Dire Dawa, Eastern Ethiopia: A cross-sectional study

**Alemu Guta** [ID] [1] *, **Habtamu Abera Areri** [2], **Kirubel Anteab** [1], **Legesse Abera** [ID] [1], **Abdurezak Umer** [1]

**1** Department of Midwifery, College of Medicine and Health Science, Dire Dawa University, Dire Dawa, Ethiopia, **2** College of Health Science, Addis Ababa University, Addis Ababa, Ethiopia

* chelsea0061@gmail.com

## Abstract

### Introduction

HIV status disclosure in children is one of acontroversial issue in current health. Over 44,000 children in Ethiopia were living with HIV in the year 2019 with a variable level of disclosure, which ranges from 16.3% to 49%. Therefore, this study aimed to assess HIV-positive status disclosure and associated factors among HIV-infected children.

### Methods

A cross-sectional study was conducted in ten public health facilities providing pediatric antiretroviral therapy services. Systematic random sampling was used to select 221 caregivers of children aged 6–15 years. Face-to-face interviews were employed to generate the data. Binary logistic regression was used to analyze the association between HIV-positive status disclosure to children and independent variables with statistical significance set at p-value <0.05.

### Results

Out of the total, 134 (60.6%) of HIV-infected children knew about their HIV status. The mean age at disclosure was 10.71 years. Children aged older than 10 years [AOR = 22, 95% CI: 5.3–79.2], female children [AOR = 3; 95% CI: 1.2–8.7], children lost their family member by HIV [AOR = 3.5, 95% CI: 1.2–10], caregiver's perception of child did not get stigmatized [AOR = 4, 95% CI: 1.6–11], and children's responsible for anti-retroviral therapy [AOR = 16, 95% CI: 5–50] were significantly associated with HIV positive status disclosure compared to their counterpart respectively. Children who stayed on anti-retroviral for 10–15 years were [AOR = 7; 95% CI: 2–27] more likely to know their HIV positive status compared to those staying on anti-retroviral therapy for <6 years.

### Conclusion

The proportion of disclosure of HIV-positive status among HIV-infected children was low. Factors associated were the age of the child, sex, existence of parent, stigma, ART duration,

**Data Availability Statement:** All relevant data are within the manuscript and Supporting Information files.

**Funding:** Dire Dawa University has supported the study with grant number: DDU/RTI/5029/2019. The University has no role in study design, data collection and analysis, decision to publish, or preparation of the manuscript.

**Competing interests:** The authors have declared that no competing interests exist.

**Abbreviations:** AIDS, Acquired Immuno Deficiency syndrome; ART, Anti-retroviral Therapy; HIV, Human Immunodeficiency Virus; MTCT, Mother to Child Transmission of HIV.

and responsibility of the child for his/her drugs. HIV care providers should consider these factors while supporting disclosure.

## Introduction

Recent estimates from UNAIDS suggest that globally, about 38 million people are living with HIV infection in 2019, about 67.3% (25.6 million) of whom live in Sub-Saharan Africa, and children younger than 15 years of age account for more than 4.7% (1.8 million) [1]. In Ethiopia, an estimated 670,000 people were living with HIV, of which children aged 0–14 years accounted for 6.5% (44,000) [1]. HIV/AIDS is increasingly affecting the health and welfare of children and undermining hard-won gains of child survival in highly affected countries [2]. Disclosure of HIV-positive status to children is controversial and emotionally charged because of caregiver fear was associated with the child may not keep secrets of their HIV diagnosis, the family secret might be disclosed outside the home, and the children get stigmatized by the community and consequences such as social rejection, both from the health care workers and parents or caregivers of the children [3–7]. However, many studies have shown that disclosure of HIV status to infected children has multiple benefits for the children and their caregivers, including psychosocial and clinical outcomes such as take responsibility for their drugs, and have improved adherence that leads to a better quality of child life [3,5,8–10].

Worldwide, of all people living with HIV, 81% knew their status [1]. According to one systematic review,in low- and middle-income countries the overall rate of HIV-positive status disclosure to HIV-infected children was low, which ranges from 1.7% to 41% [11]. Fewer studies, conducted in different cities of Ethiopia, showed that children who knew their HIV positive status were low, which ranges from 16.3% to49% [12–18]. As children mature, however, lack of disclosure may lead to accidental disclosure. Accidental disclosure from overhearing caregiver discussions may lead to both maladjustment and distrust of adults impairing treatment understanding, participation, and increase psychological and behavioral problems [19].

There are many barriers to timely disclosure of HIV-positive status to children, such as fear the child reacts to the news of the diagnosis, the age of the child, lack of caregiver readiness to discuss how the child got infected, and fear of the child on keeping secret [18].

Children with HIV disease have been called the missing face of AIDS "because, more often than adults, they lack basic health care, and they have been "missing from global and national policy discussions" [20]. The World Health Organization (WHO) recommends national policies to implement programmers with tools and resources providing clear, specific guidance on disclosure of HIV [21]. However, the process and resources available for training providers about pediatric HIV disclosure are mainly based on the Western disclosure model and experience [22,23]. They are often not adapted to low-income countries. Furthermore, studies are limited to perinatally and non-perinatally acquired HIV-positive status disclosure and associated factors among HIV-infected children aged—6–15 years in the current study area. Therefore, this study aimed to determine the prevalence of HIV-positive status disclosure and associated factors among HIV-infected children on ART treatment follow-up in Dire Dawa, Ethiopia.

## Materials and methods

### Study area, period, and design

A facility-based cross-sectional was conducted from 13th of September to 28th October 2019 in ten public health facilities providing pediatric ART services in Dire Dawa city administration, which is 515 km away from Addis Ababa to the East.

## Participants

All caregivers of HIV-positive children aged from 6 to15 years and on ART services at pediatric ART clinic of 10 health facilities (two hospitals and eight health centers) in Dire Dawa were included. However, caregivers aged <18years were excluded from the study because of ethical concerns.

## Sample size and sampling procedure

The sample size was calculated based on the following assumption: prevalence from the previous study of HIV-positive status disclosure among HIV-infected children was 16.3% [16], 95% CI, a margin of error of 0.05, and a 5% non-response rate, making a total of 221.

First, all ten public health facilities providing pediatric ART services in Dire Dawa were selected. Then, a systematic random sampling technique was used to select 221 study subjects using the sampling frame (list of HIV-infected children aged 6–15 years) was obtained from the registration book. The number of respondents was proportionally allocated to the selected health facilities, and the interview was conducted every 2$^{nd}$ interval. Finally, all eligible caregivers were interviewed during their child monthly follow-up visits at each health facility.

## Data collection, instruments, procedures, and quality assurance

Data were collected through face-to-face interviews using a pre-tested structured tool adapted from the literature. The tool included different sections such as sociodemographic characteristics, disclosure of HIV positive status, and clinical characteristics of both caregivers and children [12–16]. First, we prepared the tool in English, and then translated it to the local language and back to English by another translator to check the language consistency of the tool. Ten trained nurses from the respective health facilities were used for the collection. Two trained supervisors supervised the data collection process.

The data quality was assured by pre-testing the data collection tool on 5% (11 caregivers) of the sample size before the actual data collection. Then, corrections on the instrument were made accordingly. Each data collector checked the questionnaires for completeness before visiting the next participants. The collected data were checked for completeness every day. Then, necessary feedback was offered to the data collectors every morning.

## Measurement and definitions

The dependent variable is the disclosure of HIV-positive status to HIV-infected children (yes/no). The independent variables included (1) caregiver's related factors: age, sex, residence, marital status, religion, occupation, educational status, relation to the child, and stigma; (2) children related factors: age, sex, school grade, with whom a child lives, family existence, a child who lost a family member by HIV; and (3) clinically related factors: HIV status of caregiver, caregiver ART status, mode of transmission of HIV to the child, child age at diagnosis of HIV, WHO stage, child duration on ART, child responsibility for ART, child ART adherence, child get hospitalized, caregiver discussion about disclosure with a health care provider, and child get support from other organizations.

Disclosure refers to a child know his or her HIV positive status [3]. The primary caregiver was defined as a person who lives with the child, participates in the child's daily care, and who knows most about the child's health [14,16].

## Data processing and analysis

Data were checked for completeness and consistency. Finally, the data were entered into Epidata 3.1 and exported to SPSS version 20.0 for analysis. Frequency and percentages were used

to illustrate the study findings in tables and text. The association between the outcome and independent variables was analyzed using a binary logistic regression model. Variables with a p <0.25 in the bivariate analysis were taken for multivariable logistic regression analysis. Hosmer and Lemeshow's goodness-of-fit test was used to assess whether the necessary assumptions were fulfilled. The direction and strength of the statistical associations were measured using the odds ratio with 95% CI. AOR with 95% CI at p-value <0.05 was considered as a statistically significant association with a child know his or her HIV positive status.

### Ethical considerations

Ethical clearance was obtained from the office of director for research and technology interchange of Dire Dawa University (approval number: DDU/RTI/6076/219). Following approval, an official letter of support was given to the Region Administration Health Bureaus, hospitals, and health centers. Permission was obtained from each facility's management body. Participants were informed that participation was voluntary and that they could withdraw at any time if they were not comfortable about the questionnaire. Informed written consent was obtained from all participants. Then, the interview was carried out privately in the adherence counseling room. Names or personal identifiers were not included in the questionnaires to ensure anonymity. Furthermore, all the basic principles of human research ethics (respect for persons, beneficence, voluntary participation, confidentiality, and justice) were respected.

## Result

### Sociodemographic characteristics of caregivers and children

The response rate was 221 (100%). The majority, 165 (74.7%) and 145 (65.6%) of the caregivers were female and Christian followers by religion, respectively. Most, 211 (95.5%) and 184 (83.3%) were from urban areas and privately employed, respectively. The majority of 176 (79.6%) of the caregivers were currently married. Regarding the educational status of the caregivers, 95 (43%) attended primary school (1-8th grade). The majority 159 (,71.9%) of the children's caregivers were their mothers/fathers.

The mean age of the children was 12.1 years (SD = 3.05). The majority, 165 (74.7%), and 203 (91.9%) children were in the age range of 11–15 years and started formal education, respectively. Most, 162 (73.3%) of the children lived with their biological family, while 101 (45.7%) children lost any of their nuclear family members by HIV (**Table 1**).

### Disclosure status of HIV-positive children

Of the 221 children, 134 (60.6%, 95% CI: 54.3–67) of children infected with HIV knew their HIV status. The mean age of the child at the time of disclosure was 10.71 (SD± 2.08) years, and the majority, 95 (70.9%) of children belonged to the age range of 10–15 years. The children's HIV status was disclosed by their mothers, 84 (62.69%), health care provider, 20 (14.93%), fathers, 18 (13.43%), and grandparents, 12(9%). Of the total (134) of caregivers who responded to the reason for the disclosure, more than half 70 (52.23%) of caregivers were said the child is matured enough (**Fig 1**).

Concerning the reason for the non-disclosure, the majority of 81 (93.1%) of caregivers still believe that the child is too young, while the rest said that the child may not keep secret, fear of child self-discrimination, familychild relation may be affected, and the child may feel hopeless. Children who did not know their HIV status were told by their caregiver who they had repeated follow-up at health facilities and daily medication for TB, 24 (27.6%), allergic, 21 (24.14%), heart disease,20 (22.99%), and other diagnoses, 22 (25.3%).

**Table 1. Sociodemographic characteristics of caregiver and children in public health facilities in Dire Dawa, Eastern Ethiopia (n = 221).**

| Variables | Frequency (N) | Percentage (%) |
|---|---|---|
| **Age of caregivers** | | |
| 18–24 | 18 | 8.1 |
| 25–34 | 75 | 33.9 |
| 35–44 | 92 | 41.6 |
| 45+ | 36 | 16.3 |
| **Sex of caregivers** | | |
| Male | 56 | 25.3 |
| Female | 165 | 74.7 |
| **Religion** | | |
| Christian | 145 | 65.6 |
| Muslim | 76 | 34.4 |
| **Ethnicity** | | |
| Amhara | 116 | 52.5 |
| Oromo | 75 | 33.9 |
| Others | 30 | 13.5 |
| **Education status of the caregivers** | | |
| Non-formal education | 63 | 28.5 |
| Primary school (1-8th) | 95 | 43 |
| 2ndary and above | 63 | 28.5 |
| **Occupation** | | |
| Governmental employed | 37 | 16.7 |
| Private worker | 184 | 83.3 |
| **Marital status** | | |
| Single | 22 | 10.0 |
| Married | 176 | 79.6 |
| Divorced | 11 | 5.0 |
| Widowed | 12 | 5.4 |
| **Relationship to child** | | |
| Biological caregivers | 159 | 71.9 |
| Non-biological caregivers | 62 | 28.1 |
| **Age of the child** | | |
| 6–10 years | 56 | 25.3 |
| 11–15 years | 165 | 74.7 |
| **Sex of the child** | | |
| Male | 126 | 57.0 |
| Female | 95 | 43.0 |
| **Child school grade** | | |
| Not started education | 18 | 8.1 |
| Start formal education | 203 | 91.9 |
| **A child lost any family member by HIV** | | |
| Yes | 101 | 45.7 |
| No | 120 | 54.3 |
| **Parent/s lost** | | |
| Mother only | 30 | 13.6 |
| Father only | 39 | 17.6 |
| Both mother and father | 32 | 14.5 |

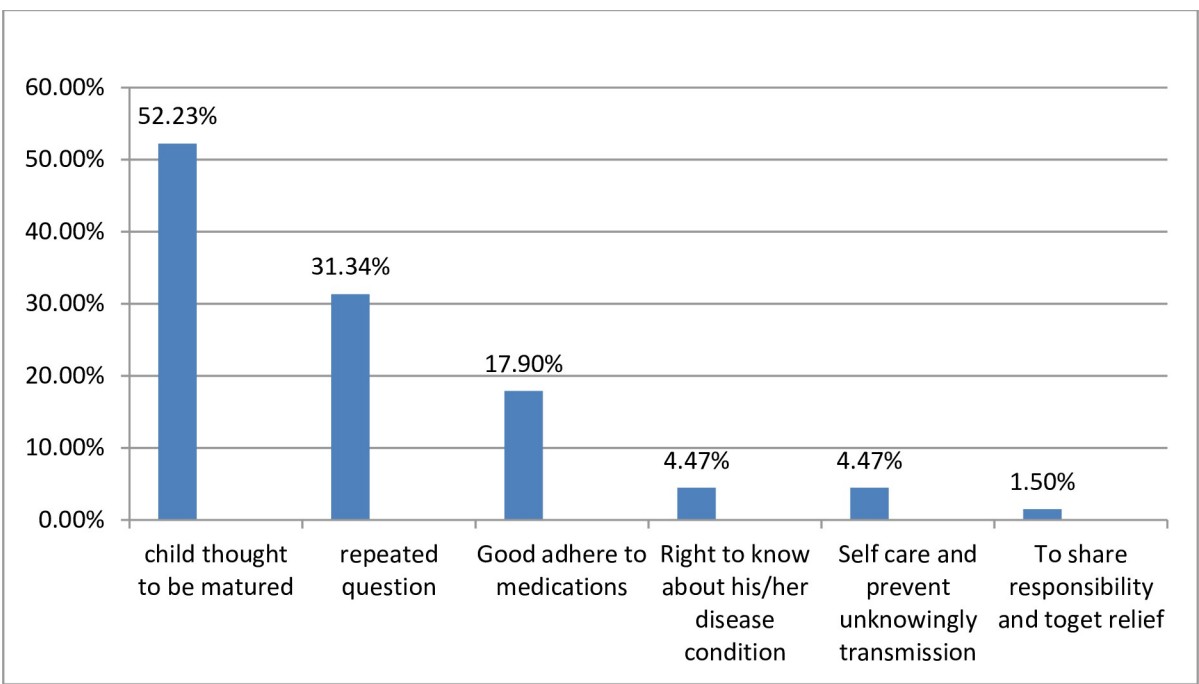

**Fig 1. Caregivers reason for disclosing HIV-positive status to children LHIV+.**

"Majority 73 (83.9%) of the caregivers had the plan to disclose in the future while the rest did not have a plan. Concerning the preferred age of disclosure, the majority, 158 (71.5%) were said 10–15 years of age, and the remaining 16.3%, and 5.9% of caregivers said 6–9 years, and above 15years, and only 6.3% of the caregivers not sure. The majority believed that mothers, 157 (71%) health care providers, 85 (38.5%), and fathers, 23 (10.4%) were responsible for disclosing the HIV status of the children. Over fifty percent 121 (54.8%) of caregivers replied that their child is did not get stigmatized due to their HIV-positive status.

## Clinical characteristics of children and caregivers

The majority 204 (;92.3%) of the caregivers were HIV-positive. Most, 219 (99.1%) of the respondents replied that the mode of transmission of HIV to the child was perinatal. Greater than three-fourth (78.3%) of the children were diagnosed under five years of age and the mean age at diagnosis was 3.31 years (SD = 2.67). The mean age of children for the duration of ART was 8.82 years (SD = 3.7), and the majority, 193 (87.3%) children were on good treatment adherence status. The majority 200 (90.5%) of the children had a WHO clinical stage I disease, and 90 (40.7%) were admitted to the hospital previously. The majority 196 (88.7%) participants discussed disclosure issues with their child's health care provider, and most of them got adequate information about disclosure. Only 48(21.7%) of the children got a variety of support from different organizations, including money, 36(16.3%) food, 7(3.2%), and counseling 5(2.3%) (**Table 2**).

## Factors with HIV-positive status disclosure

In this study, 12 variables were significantly associated with bivariate analysis. Six variables were significantly associated with disclosure in the multiple logistic regression analysis models (p <0.05). These factors included the following: age of the child, sex of the child, child loss of a family member by HIV, caregiver's perception of child stigmatized, child responsible for ART, and ART duration

**Table 2. Clinical characteristics of caregivers and children in public health facilities in Dire Dawa, 2019 (n = 221).**

| Variables | Frequency (N) | Percent (%) |
|---|---|---|
| **HIV status of the caregiver** | | |
| Positive | 204 | 92.3 |
| Negative | 17 | 7.7 |
| **Child age at diagnosis Mean+SD** | | 3.31 +2.67 |
| **Age at diagnosis of HIV** | | |
| <6 years | 187 | 84.6 |
| 6–9 years | 20 | 9.0 |
| 10–15 years | 14 | 6.3 |
| **WHO clinical stage** | | |
| Stage I | 200 | 90.5 |
| Stage II | 21 | 9.5 |
| **Duration on the child on ART Mean+SD** | | 8.82+3.7 |
| **Duration child on ART** | | |
| <6 year | 187 | 84.6 |
| 6–9 year | 20 | 9.0 |
| 10–15 year | 14 | 6.3 |
| **Child responsible for ART** | | |
| No | 136 | 61.5 |
| Yes | 85 | 38.5 |
| **Treatment adherence of the child** | | |
| Good | 193 | 87.3 |
| Fair | 28 | 12.7 |
| **Child hospital admitted previously** | | |
| Yes | 90 | 40.7 |
| No | 131 | 59.3 |
| **Caregivers discuss disclosure** | | |
| Yes | 196 | 88.7 |
| No | 25 | 11.3 |
| **A child got support from other organizations** | | |
| Yes | 48 | 21.7 |
| No | 173 | 78.3 |

of the child. Those children aged older than ten years were 22 times more likely to know their HIV positive status compared to their counterparts [AOR = 22, 95% CI: 5.3–79.2]. Female children were 3 times more likely to be disclosed about their HIV status compared to their counterparts [AOR = 3; 95% CI = 1.2–8.7]. Those children who lost any of their family members by HIV were 3.5 times more likely to disclose their HIV-positive status compared to their counterparts [AOR = 3.5, 95% CI: 1.2–10]. Caregivers who perceived that their child did not get stigmatized due to their HIV positive status were 4 times [AOR = 4, 95% CI: 1.6–11] more likely to disclose their child HIV-positive status compared to their counterparts. Similarly, those children who stayed on ARV drugs for 10–15 years were 7 times [AOR = 7; 95% CI: 2–27] more likely to know about their HIV status compared to their counterparts. (**Table 3**).

## Discussion

This study aimed to determine the prevalence of HIV-positive status disclosure and associated factors among HIV-positive children aged between 6 and 15 years. The study findings revealed

**Table 3. Factors associated with HIV positive status disclosure among HIV infected children in public health facilities in Dire Dawa, 2019 (n = 221).**

| Variables | Disclosure status (N = 221) | | COR (95%CI) | AOR (95%CI) | P-value |
|---|---|---|---|---|---|
| | Yes (N = 134) | No (N = 87) | | | |
| **Sex of the child** | | | | | |
| Male | 65 | 61 | 1 | 1 | |
| Female | 69 | 26 | 2.5(1.4–4.4) | 3 (1.2–8.7) | 0.026 |
| **Child age** | | | | | |
| = <10 year | 6 | 50 | 1 | 1 | |
| >10 year | 128 | 37 | 29 (11.5–72) | 22(5.3–79.2) | 0.000* |
| **Loss of family member by HIV** | | | | | |
| No | 64 | 56 | 1 | 1 | |
| Yes | 70 | 31 | 2 (1.14–3.4) | 3.5 (1.2–10) | 0.020* |
| **Caregiver relation (type)** | | | | | |
| Biological parent | 92 | 67 | 1 | 1 | |
| Non-biological | 42 | 20 | 1.5(0.8–2.8) | 1.1(0.3–4) | 0.918 |
| **Child gets stigmatized** | | | | | |
| No | 90 | 31 | 3.7 (2.1–6.5) | 4 (1.6–11) | 0.004* |
| Yes | 44 | 56 | 1 | 1 | |
| **WHO clinical stage** | | | | | |
| Stage 1 | 126 | 74 | 2.7(1–7) | 2 (0.4–9) | 0.427 |
| Stage 2 | 8 | 13 | 1 | 1 | |
| **Child duration on ART** | | | | | |
| 0–5 year | 10 | 34 | 1 | 1 | |
| 6-9year | 26 | 36 | 2.5 (1–5.8) | 1.6(0.4–6.5) | 0.529 |
| 10-15year | 96 | 17 | 19 (8–45) | 7(1.8–27) | 0.005* |
| **Child responsible for medication** | | | | | |
| No | 61 | 75 | 1 | 1 | |
| Yes | 73 | 12 | 7.5 (3.7–15) | 16 (5–50) | 0.000* |
| **Child treatment adherence** | | | | | |
| Good | 120 | 73 | 1.6(0.7–3.6) | 3.4 (0.8–13) | 0.085 |
| Fair | 14 | 14 | 1 | 1 | |
| **History of hospitalization** | | | | | |
| No | 69 | 62 | 1 | 1 | |
| Yes | 65 | 25 | 2.3 (1.3–4) | 2(0.27–2.5) | 0.572 |
| **Discuss with HCP** | | | | | |
| No | 7 | 18 | 1 | 1 | |
| Yes | 127 | 69 | 4.7 (2–12) | 6 (1–39) | 0.056 |
| **Support from other organization** | | | | | |
| No | 97 | 76 | 1 | 1 | |
| Yes | 37 | 11 | 2.6 (1.3–5.5) | 2(0.63–7) | 0.230 |

*Statistically significant at p-value <0.05 in multivariate logistic regression analysis.

that 60.6% of HIV-infected children knew their HIV status. This finding is lower than those reported in the U.S.A and Canada, which were 75–82% [6,24]. The finding of this study is slightly similar to the survey conducted in Uganda, which is (56%) [25]. However, this finding is higher than studies conducted in South Africa, Ghana, CotedIvoire, and Kenya showed that the disclosure rates were 34%, 21%, 32.6%, 56%, and 10.2%, respectively [26–29]. Similarly, this finding is higher than studies conducted in three cities in Gondar in 2012 and 2018, Bahir Dar and Debre Markos, which were 39.5%, 44, 31.5%, and 33.3%, respectively [12–14,18]. This

difference might be due to the caregivers believed that disclosing the HIV-positive status to the child has an advantage and children have to disclose their status to have a long-term medical follow-up. HIV care providers should encourage the disclosure status of children during follow-up visits and should discuss with caregivers the possible way of disclosure.

On the other hand, the finding is higher compared to two studies conducted in Addis Ababa, which reported a disclosure rate of 17.4% and 16.3% [15,16]. The possible justification for the discrepancy might be a cultural difference, the elapsed study period, and the difference in awareness about disclosure by the caregivers. Besides, the study in Addis Ababa, unlike the current study, assessed the disclosure rate of all pediatric age. Besides, most of the caregivers in the current study attended primary education and discussed disclosure issues with children's health care providers. However, it is expected that most children should know their HIV status to minimize the psychological and social impact of the non-disclosure on the later life of the children.

In our study, children aged older than 10 years were 22 times more likely to be disclosed as knew their HIV positive status compared to their counterparts. is this finding supported by studies conducted in Addis Ababa in 2012, Gonder 2019, East Gojjam, Ghana, South Africa, and Uganda [14,16,18,27,30]. The possible justification might be thatthe caregivers believed that the child was mature enough to understand the illness. In addition, they can keep a secret of HIV disclosure, accept their disease condition, and take responsibility for his/her case.

The odds of disclosure of HIV-positive status among children who are staying on ART for 10–15 years were 7 times more likely to disclose their HIV positive status than those who take ART medication less than six-year. This finding is consistent with studies conducted in Bahir Dar, Gondar, and Ghana [13,18,27]. This could be explained that when children had been on ART for a while, they no longer experienced symptoms. Therefore, they asked their caregivers' subsequent questions about their HIV medications, which resulted in a lack of adherence to treatment, and finally, that leads to disclosure. This calls HIV care providers to work with caregivers to clarify uncertainty closely.

The child responsible for his/her medication is also another factor significantly associated with HIV positive status disclosure. This finding was in line with the studies done in Ghana and East Gojjam [14,27]. This is due to disclosure affects drug adherence, and it is possible that after disclosure, children took charge of their illness and medication, leading to improved adherence.

This study has some limitations. The caregiver reports of HIV status disclosure could be affected by social desirability bias, and they might have said that they disclosed when they had not indeed disclosed.

## Conclusion

The prevalence of disclosure of HIV-positive status to HIV-infected children was low. Factors associated were the age of the child, gender, existence of parents, stigma, ART duration, and responsibility of the child for his/her drugs. HIV care providers should consider these factors while working with caregivers to encourage disclosure of HIV-positive status. For a better understanding of the reasons for non-disclosure qualitative study is recommended for future study.

## Supporting information

**S1 Questionnaire. The data collection tool in the local language, Amharic.**
(DOCX)

**S2 Questionnaire. The data collection tool in English.**
(DOCX)

**S1 Dataset. The dataset from which the results of the study were produced (SPSS file).**
(SAV)

**S1 Certificate. PubSURE English editing certificate.**
(PDF)

## Acknowledgments

First and for most, we would like to thank Dire Dawa University Research and TechnologyInterchange, College of Medicine and Health Sciences, and all respective Health Bureau for their support and collaborations. Secondly, we would like to thank all data collectors and supervisors for their valuable contribution during the data collection. Lastly, we would like to thank the study participants.

## Author Contributions

**Conceptualization:** Alemu Guta.

**Data curation:** Alemu Guta.

**Formal analysis:** Alemu Guta, Habtamu Abera Areri, Kirubel Anteab.

**Investigation:** Alemu Guta.

**Methodology:** Alemu Guta, Kirubel Anteab, Legesse Abera, Abdurezak Umer.

**Project administration:** Alemu Guta.

**Resources:** Alemu Guta.

**Software:** Alemu Guta.

**Supervision:** Alemu Guta.

**Validation:** Alemu Guta.

**Visualization:** Alemu Guta.

**Writing – original draft:** Alemu Guta.

**Writing – review & editing:** Alemu Guta, Habtamu Abera Areri, Kirubel Anteab, Legesse Abera, Abdurezak Umer.

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
