## [Decision Letter · Decision Letter 0]

28 Jul 2020

PONE-D-20-07455

HIV-positive disclosure and associated factors among children in public health facilities in Dire Dawa, Eastern Ethiopia.

PLOS ONE

Dear Dr. Guta,

Thank you for submitting your manuscript to PLOS ONE. After careful consideration, we feel that it has merit but does not fully meet PLOS ONE’s publication criteria as it currently stands. Therefore, we invite you to submit a revised version of the manuscript that addresses the points raised during the review process.

The manuscript has been evaluated by two reviewers, their comments are available below.

The reviewers have raised a number of concerns, regarding the reporting of methodological aspects of the study such as inclusion/exclusion criteria, and data abstraction. In addition, they request a careful consideration of the discussion to ensure that the relevance of this study is further contextualised. Please also note that your manuscript requires copyediting.

Could you please carefully revise the manuscript to address all comments raised.

We look forward to receiving your revised manuscript.

Kind regards,

Sara Fuentes Perez, PhD

Staff Editor

PLOS ONE

Journal Requirements:

Additional Editor Comments (if provided):

Reviewers' comments:

Reviewer's Responses to Questions

**Comments to the Author**

1. Is the manuscript technically sound, and do the data support the conclusions?

Reviewer #1: Yes

Reviewer #2: Yes

2. Has the statistical analysis been performed appropriately and rigorously? 

Reviewer #1: Yes

Reviewer #2: I Don't Know

3. Have the authors made all data underlying the findings in their manuscript fully available?

Reviewer #1: Yes

Reviewer #2: Yes

4. Is the manuscript presented in an intelligible fashion and written in standard English?

Reviewer #1: Yes

Reviewer #2: No

5. Review Comments to the Author

Reviewer #1: The manuscript is suitable for publication considering the methodological rigor in conducting the study, which implied the presentation of advances in the knowledge of the theme, which is current and relevant to the health area. I suggest informing that the study included children and adolescents. This may reinforce the result of the disclosure taking place during childhood.

Reviewer #2: Disclosure of HIV status to children is an important topic of study.

The authors should look at literature from sub-Saharan Africa more widely, including on interventions already proven effective and being implemented at scale in the region.

The grammar and punctuation used throughout the paper is a concern and should be improved on for readers to benefit from this work

Abstract

Intro: 2.12% - A really precise measure that is very unlikely

-State the actual prevalence instead of mentioning that disclosure is low

Methods: Cross sectional study design rather than data collection.

-Mention number of facilities and sampling method for participants rather than just saying it was random

-Face to face interviews? - in-depth interviews or just questionnaires?

-Logistic regression to answer which question?

Results: Rectify use of word rate to mean proportion or prevalence

-State words in full before abbreviations - AOR, ART

-AOR of 22 with 95%CI of 5.3-9.2 erroneous

-State variable clearly -example: What does "children lost their family" mean?

-In presenting results of the regression, mention the comparator like you did in the last line of the results

Main text

Intro

-line 63: Starts with even though but does not flow logically

-Why do the authors feel that disclosure is controversial? Emotional yes! but there is hardly any controversy on the importance of disclosure. Maybe when or who should disclose? Justify

-The rationale of the study is not well elaborated in the intro

- There are multiple studies in SSA and even Ethiopia, why should we read another paper on disclosure? What is the gap in literature you what to fill? What is the expected impact of your findings?

Methods

-State clearly - Study design (cross sectional), conducted in ..... number of facilities

-You mention 1106 children. Were they from the 11 facilities where the study took place? What is the relevance of this information?

-Streamline the inclusion criteria e.g. targeted caregivers of children 6-15; excluded a,b,c

Note: Is it possible that excluding children who were unaccompanied means excluding a big population that is disclosed to?

-Later in the methods you mention that sampling was done using registration books. Where does the inclusion criteria and this sampling method meet?

-What is the difference between line 98 and 99? child came to clinic alone in both?

-Line 102 - was p value used in place of prevalence? Was the study cited done in the same area?

-Line 122 - MOT?

-What data was abstracted from clinic records?

Analysis

-What is the question being answered by the regression analysis?... If you want to assess factors associated with disclosure say so then state the dependent and independent variables being assessed

-Why use p value of 0.25 to select variables for the multivariable analysis?

Results

-Sampled 221, approached 221 and 221 responded to questions?

-Standardize presentation of results - the use of fractions, counts and proportions all at once is atypical. Standard presentation: count (proportion/percentage)

Couple means they were married?

-Line 172: State results clearly - Of the XXX not disclosed to, YYY said....

-Line 178: CHF?

-Consider arranging tables more logically - maybe table 1: Caregiver and child socio-demographic and HIV treatment history, table 2: Disclosure characteristics

-Line 201 -p value used now 0.2? Further, this is not a measure of significance but criteria for inclusion in the multivariable analysis?

-Again AOR of 22 and 95%CI of up to 9.2??

Discussion

-Line 233 to 236 - Are there studies that inform this opinion?

-Focus your discussion on the implication of your findings and not just how they compare with other studies

Conclusion

-You state that the prevalence of disclosure is low, yet higher than most places in Africa??

6. PLOS authors have the option to publish the peer review history of their article (what does this mean?). If published, this will include your full peer review and any attached files.

Reviewer #1: **Yes: **Stela Maris de Mello Padoin

Reviewer #2: **Yes: **Cyrus Mugo

---

## [Author Response · Author response to Decision Letter 0]

26 Aug 2020

Date, August 26, 2020________________________________________

To: PLOS one, editor-in-chief 

Subject: Submitting response to reviewers comment [PONE-D-20-07455] 

We thank the editor and reviewers for evaluating our manuscript entitled “HIV-positive status disclosure and associated factors among children in public health facilities in Dire Dawa, Eastern Ethiopia”. We appreciate the reviewers and editor comment on the previous version of the manuscript and have responded to these in the revised version. In this document, we have described all changes made following the comments of reviewers and editor. Our responses are given in a point-by-point manner below. Follow our responses in green font mark. All changes stated here in response to reviewers’ are incorporated in the revised manuscript. 

With Kind regards 

Comment:

Response: Thank you for your comment, we accept your comment and correction was made based on the PLOS One formatting.

Comment:

Whilst you may use any professional scientific editing service of your choice, PLOS has partnered with both American Journal Experts (AJE) and Edit age to provide discounted services to PLOS authors. Both organizations have experience helping authors meet PLOS guidelines and can provide language editing, translation, manuscript formatting, and figure formatting to ensure your manuscript meets our submission guidelines. To take advantage of our partnership with AJE, visit the AJE website (http://learn.aje.com/plos/) for a 15% discount off AJE services. To take advantage of our partnership with Editage, visit the Editage website (www.editage.com) and enter referral code PLOSEDIT for a 15% discount off Editage services. If the PLOS editorial team finds any language issues in text that either AJE or Editage has edited, the service provider will re-edit the text for free.

Response: We thank you for advising us to make editions to the writing in English regarding this manuscript. We used “grammarly” online software to edit the spelling, grammar and language usage. Following that grammarly-online suggested there are word choice issues we approached PubSURE a research paper language editing service. Moreover, we used a PubSURE software and we secured a certificate of quality of language that has pass score of 97%. We have uploaded the certificate as supporting information. Regarding use of professional editing, we don’t afford to pay the required payment. Moreover, currently we have made changes in the manuscript in track changes and uploaded as supporting files. We uploaded a clean copy of the edited manuscript with file name *manuscript* file.

https://bmcpediatr.biomedcentral.com/articles/10.1186/s12887-018-1330-5

https://www.ncbi.nlm.nih.gov/pubmed/23782475

https://www.hilarispublisher.com/open-access/challenges-of-caregivers-to-disclose-their-childrens-hiv-positive-status-receiving-highly-active-anti-retroviral-therapy-at-pediatric-anti-retroviral-therapy-clinics-2155-6113.1000253.pdf

In your revision ensure you cite all your sources (including your own works), and quote or rephrase any duplicated text outside the methods section. Further consideration is dependent on these concerns being addressed.

Response: We thank you for your comment, we revised and made all citation properly, and rephrashe any duplicated text with those of three previous studies conducted in Bahir Dar, Ghana and Coˆte d’Ivoire. We incorporated the correction throughout the revised manuscript.

Comment:

4. Please include additional information regarding the survey or questionnaire used in the study and ensure that you have provided sufficient details that others could replicate the analyses. For instance, if you developed a questionnaire as part of this study and it is not under a copyright more restrictive than CC-BY, please include a copy, in both the original language and English, as Supporting Information.

Please also state whether you validated the questionnaire prior to testing on study participants. Please provide details regarding the validation group within the methods section.

Response: Thank you, indeed for the advices on the revision. We have attached the questionnaires in both the local language (Amharic) and English we used for the current survey. Please, check in the supporting files. We pre-tested the material on 5% (11 of caregivers of children) of samples and modified the tool. Moreover, we used the questionnaire from previous literature, we did not follow standard validation procedure than pre-testing. Based on the pretest findings we made the following improvement to the questionnaire:

As much as possible the length of the questions were made shorter

Instructions for how to ask and responded were added or improved to every important section of the questionnaire

Comment:

Response: Thank you for your comment, we included the information about financial disclosure as ‘Dire Dawa University has supported the study with grant number: DDU/RTI/5029/2019. The University has no role in study design, data collection and analysis, decision to publish, or preparation of the manuscript.’

Comment:

6. Please upload a copy of Supporting Information which you refer to in your text on page 19.

Response: Thank you, we have attached the SPSS file. Check in the supplementary files.

Comment:

7. Please amend your current ethics statement to confirm that your named ethics committee Institutional Care and Use Committee (IACUC) specifically approved this study.

Response: Thank you your comment, we have incorporqated the name the institution that provide us the ethical clearance in the manuscript as ‘Ethical clearance was obtained from the office of director for research and technology interchange of Dire Dawa University (approval number: DDU/RTI/6076/219).’ 

Comment:

8. Thank you for updating your data availability statement. You note that your data are available within the Supporting Information files, but no such files have been included with your submission. At this time we ask that you please upload your minimal data set as a Supporting Information file, or to a public repository such as Figshare or Dryad. Please also ensure that when you upload your file you include separate captions for your supplementary files at the end of your manuscript. As soon as you confirm the location of the data underlying your findings, we will be able to proceed with the review of your submission.

Response: Thank for your comment; we have uploaded the supppourtive files as suppourtive information now.

Comment:

The reviewers have raised a number of concerns, regarding the reporting of methodological aspects of the study, such as inclusion/exclusion criteria, and data abstraction. In addition, they request a careful consideration of the discussion to ensure that the relevance of this study is further contextualized. Please also note that your manuscript requires copyediting.

Response: Thank for your comment; we have carefully reviewed and made the correction as required, all our correction is incorporated into the revised manuscript. 

Thank you for your comment, English language copy-editing acted.

Could you please carefully revise the manuscript to address all comments raised.

Response: Thank you for your comment; yes, we acted on the commented and incorporated the correction throughout the revised manuscript.

4. Is the manuscript presented in an intelligible fashion and written in Standard English?

Reviewer #1: Yes

Reviewer #2: No

Response: English language copy-editing done throughout the paper.

5. Review Comments to the Author

Reviewer #1: The manuscript is suitable for publication considering the methodological rigor in conducting the study, which implied the presentation of advances in the knowledge of the theme, which is current and relevant to the health area. I suggest informing that the study included children and adolescents. This may reinforce the result of the disclosure taking place during childhood.

Reviewer #2: Disclosure of HIV status to children is an important topic of study.

The authors should look at literature from sub-Saharan Africa more widely, including on interventions already proven effective and being implemented at scale in the region.

Comment; The grammar and punctuation used throughout the paper is a concern and should be improved on for readers to benefit from this work

Response: Thank you for your comment, we revised the grammar and punctuation used throughout the paper, and the correction is incorporated the revised manuscript.

Abstract

Comment; Intro: 2.12% - A really precise measure that is very unlikely

-State the actual prevalence instead of mentioning that disclosure is low

Response: Thank you for your comment; we accept the comment and both the number of …children living with HIV and the actual prevalence of HIV positive status disclosure stated as; Over 44,000 children in Ethiopia were living with HIV in the year 2019 with a variable level of disclosure, which ranges from 16.3% to 49%.The correction is incorporated the revised manuscript.

Comment; Methods: Cross sectional study design rather than data collection.

Response: The comment is accepted and corrected; A cross-sectional study was conducted.

Comment; -Mention number of facilities and sampling method for participants rather than just saying it was random

Response: The comment is accepted, and the ten health facilities and the sampling techniques’ were revised and rewritten as systematic random sampling was used to select 221 caregivers of children aged 6-15 years.

Comment; -Face to face interviews? - In-depth interviews or just questionnaires?

Response: Face to face interviews? yes, it is face-to-face interview questionnaire 

Comment; -Logistic regression to answer which question?

Response: We accepted your comment and corrected logistic regression analysis as binary logistic regression was used to analyze the association between HIV-positive status disclosure to children and independent variables with statistical significance set at p-value <0.05.

Comment; Results: Rectify use of word rate to mean proportion or prevalence

Response: Thank you for your comment; we accepted your comment and rate changed to proportion in the revised manuscript

Comment; -State words in full before abbreviations - AOR, ART

Response: Thank you for your comment; we accepted your comment and correction incorporated in the revised manuscript.

Comment; -AOR of 22 with 95% CI of 5.3-9.2 erroneous

Response: Thank you for your comment; we accepted your comment and our typing erroneous of the CI, which is corrected as 5.3-79.2.

Comment; -State variable clearly -example: What does "children lost their family" mean?

Response: We accept the comment and corrected, children lost their family member means children lost their family member by HIV

Comment; -In presenting results of the regression, mention the comparator like you did in the last line of the results

Response: Thank you for your comment; we accepted your comment and we try to interpreted those 5 significantly associated variables as following: children aged older than 10 years [AOR = 22, 95% CI: 5.3- 59.2], female children [AOR = 3; 95% CI = 1.2-8.7], children lost their family member by HIV [AOR = 3.5, 95% CI: 1.2-10], caregiver’s perception of child did not get stigmatized due to his/her HIV positive status [AOR = 4, 95% CI: 1.6-11], and children’s responsible for ART [AOR = 16, 95% CI: 5-50] were significantly association with HIV positive status compared to their counterpart respectively.

But, if the 5 variables are interpreted one-by-one, it raises the number of word count in the abstract. 

Main text

Comment; Intro

-line 63: Starts with even though but does not flow logically

-Why do the authors feel that disclosure is controversial? Emotional yes! but there is hardly any controversy on the importance of disclosure. Maybe when or who should disclose? Justify

Response: Thank you for your comment; we accepted your comment and rewritten the paragraph on line 63-66 or which started with even though most children infected with HIV…. We incorporated the modified sentence into the revised manuscript.

Comment; -The rationale of the study is not well elaborated in the intro

- There are multiple studies in SSA and even Ethiopia, why should we read another paper on disclosure? What is the gap in literature you what to fill? What is the expected impact of your findings?

Response: Thank you for your comment; we accepted the comment, and we added more elaboration to the rationale of the study, the gap and expected impact of our study for different stakeholders in the revised manuscript… Furthermore, studies are limited to perinatally and non-perinatally acquired HIV-positive status disclosure and associated factors among HIV-infected children aged -- 6-15 years in the current study area. Therefore, this study aimed to determine the prevalence of HIV-positive status disclosure and associated factors among HIV-infected children on ART treatment follow-up in Dire Dawa, Ethiopia. This correction is also incorporated in the revised manuscript.

Methods

Comment; -State clearly - Study design (cross-sectional), conducted in ..... number of facilities

Response: Thank you for your comment; we accepted the comment and corrected as 10 health facilities

Comment; -You mention 1106 children. Were they from the 11 facilities where the study took place? What is the relevance of this information?

Response: The comment is accepted, and we removed the number of children mentioned

Comment; -Streamline the inclusion criteria e.g. targeted caregivers of children 6-15; excluded a,b,c

Note: Is it possible that excluding children who were unaccompanied means excluding a big population that is disclosed to?

Response: Thank you for your comment; 

We corrected as; the inclusion criteria is all caregivers of HIV-positive children aged from 6 to15 years and on ART services at pediatric ART clinic of 10 health facilities (two hospitals and eight health centers) in Dire Dawa were included.

As exclusion caregivers aged <18years were excluded from the study because of ethical concerns.

By default, child come alone is excluded, they are not our study population, besides from an ethical point of view we could not interview under 18 years without caregivers consent.

Comment; -Later in the methods you mention that sampling was done using registration books. Where does the inclusion criteria and this sampling method meet?

Response: Thank you for your comment; we revised the sentence; we used the registration book to know a number of children, but the data collection is employed at the time of child monthly follow-up visit.

Comment; -What is the difference between line 98 and 99? Child came to clinic alone in both?

Response: Thank you for your comment; we accepted the comment and corrected.

Comment; -Line 102 - was p value used in place of prevalence? Was the study cited done in the same area?

Response: Thank you for your comment; we accepted the comment, and the p-value is changed to prevalence.

Comment; -Line 122 - MOT?

Response: Thank you for your comment; MOT stands for a mode of transmission

Comment; -What data was abstracted from clinic records?

Response: Thank you for your comment; we revised the child clinical record means, of course, the information addressed by child clinical record is also in the interview questionnaire; we used child clinical record concomitantly with interview questionnaire to fill the gap that raised during face to face interview questionnaires. Example, the question on clinical characteristics of child WHO clinical stage, age at diagnosis

B/c, the interview was employed during the monthly follow-up visit of the children.

As it is supportive, we are removed the sentence.

Analysis

Comment; -What is the question being answered by the regression analysis? If you want to assess factors associated with disclosure say so then state the dependent and independent variables being assessed

Response: Thank you for your comment; we accepted the comment and incorporated the correction in the revised manuscript as follow; The dependent variable is the disclosure of HIV-positive status to HIV-infected children (yes/no). The independent variables included (1) caregiver’s related factors: age, sex, residence, marital status, religion, occupation, educational status, relation to the child, and stigma; (2) children related factors: age, sex, school grade, with whom a child lives, family existence, a child who lost a family member by HIV; and (3) clinically related factors: HIV status of caregiver, caregiver ART status, mode of transmission of HIV to the child, child age at diagnosis of HIV, WHO stage, child duration on ART, child responsibility for ART, child ART adherence, child get hospitalized, caregiver discussion about disclosure with a health care provider, and child get support from other organizations.

Comment; -Why use p value of 0.25 to select variables for the multivariable analysis?

Response: Thank you for your comment; It is not to miss confounder and possibly to improve the strength of the model.

Results

Comment; -Sampled 221, approached 221 and 221 responded to questions?

Response: Thank you for your comment; yes, the sampled caregivers were interviewed.

Comment; -Standardize presentation of results - the use of fractions, counts and proportions all at once is atypical. Standard presentation: count (proportion/percentage)

Response: Thank you for your comment; we accepted the comment and corrected as a standard presentation of the results: count (percentage) and the correction is incorporated inthe revised manuscript.

Comment; Couple means they were married?

Response: Thank you for your comment; accepted, and the couple is changed to married.

Comment; -Line 172: State results clearly - Of the XXX not disclosed to, YYY said....

Response: Thank you for your comment; the comment is accepted, and the sentence clearly is written, and the correction is included in the revised manuscript.

Comment; -Line 178: CHF?

Response: Thank you for your comment; CHF stands for congestive heart failure (ask as heart disease on the questionnaire)

Comment; -Consider arranging tables more logically - maybe table 1: Caregiver and child socio-demographic and HIV treatment history, table 2: Disclosure characteristics

Response: The comment is accepted, and we logical arranged our tables.

Comment; -Line 201 -p value used now 0.2? Further, this is not a measure of significance but criteria for inclusion in the multivariable analysis?

Response: Thank you for your comment; we accept the comment and corrected

Comment; -Again AOR of 22 and 95%CI of up to 9.2??

Response: Thank you for your comment; we accept the comment and corrected our typing erroneous of the CI, which is corrected by 5.3-79.2

Discussion

Comment; -Line 233 to 236 - Are there studies that inform this opinion?

-Focus your discussion on the implication of your findings and not just how they compare with other studies

Response: Thank you for the comment; yes, we observed the gap, and we acted on the commented and incorporated the correction throughout the revised manuscript. 

Conclusion

Comment; -You state that the prevalence of disclosure is low, yet higher than most places in Africa??

Response: Thank you for the comment; our study is recent compared to those done before where issues related to HIV is serious compared to now. In addition, compared to the interventions given to enhance the level of disclosure, it is considered low.

---

## [Decision Letter · Decision Letter 1]

14 Sep 2020

HIV-positive status disclosure and associated factors among children in public health facilities in Dire Dawa, Eastern Ethiopia. A cross-sectional study

PONE-D-20-07455R1

Dear Mr Guta,

We’re pleased to inform you that your manuscript has been judged scientifically suitable for publication and will be formally accepted for publication once it meets all outstanding technical requirements.

Kind regards,

Joseph K.B. Matovu, Ph.D.

Academic Editor

PLOS ONE

Additional Editor Comments (optional):

Reviewers' comments:

Reviewer's Responses to Questions

**Comments to the Author**

1. If the authors have adequately addressed your comments raised in a previous round of review and you feel that this manuscript is now acceptable for publication, you may indicate that here to bypass the “Comments to the Author” section, enter your conflict of interest statement in the “Confidential to Editor” section, and submit your "Accept" recommendation.

Reviewer #1: All comments have been addressed

Reviewer #2: All comments have been addressed

2. Is the manuscript technically sound, and do the data support the conclusions?

Reviewer #1: Yes

Reviewer #2: Yes

3. Has the statistical analysis been performed appropriately and rigorously? 

Reviewer #1: Yes

Reviewer #2: Yes

4. Have the authors made all data underlying the findings in their manuscript fully available?

Reviewer #1: Yes

Reviewer #2: Yes

5. Is the manuscript presented in an intelligible fashion and written in standard English?

Reviewer #1: Yes

Reviewer #2: Yes

6. Review Comments to the Author

Reviewer #1: The manuscript met the comments of the reviewers and editor and is suitable for publication, the topic is current and relevant to the area of health.

Reviewer #2: The authors have addressed my comments. The few typos noted can be corrected in the proofing stage before publication.

7. PLOS authors have the option to publish the peer review history of their article (what does this mean?). If published, this will include your full peer review and any attached files.

Reviewer #1: **Yes: **Stela Maris de Mello Padoin

Reviewer #2: **Yes: **Cyrus Mugo

---

## [Editor Report · Acceptance letter]

30 Sep 2020

PONE-D-20-07455R1 

HIV-positive status disclosure and associated factors among children in public health facilities in Dire Dawa, Eastern Ethiopia: A cross-sectional study

Dear Dr. Guta:

I'm pleased to inform you that your manuscript has been deemed suitable for publication in PLOS ONE. Congratulations! Your manuscript is now with our production department. 

Kind regards, 

on behalf of

Dr. Joseph K.B. Matovu 

Academic Editor

PLOS ONE